# Exploring Language Recovery Pattern in Persons with Aphasia Across Acute and Sub-Acute Stages

**DOI:** 10.3390/bs15101339

**Published:** 2025-09-29

**Authors:** Deepak Puttanna, Nova Maria Saji, Mohammed F. ALHarbi, Akshaya Swamy, Darshan Hosaholalu Sarvajna

**Affiliations:** 1Department of Audiology and Speech Language Pathology, College of Medical Rehabilitation Sciences, Taibah University, Al Medina 42353, Saudi Arabia; mfmharbi@taibahu.edu.sa (M.F.A.); akshuswamy@gmail.com (A.S.); 2Let’s Talk Speech Therapy Center, Mumbai 400004, India; novasaji99@gmail.com; 3Nitte (Deemed to be University), Nitte Institute of Speech and Hearing (NISH), Mangalore 575018, India; darshan.hs@nitte.edu.in

**Keywords:** stroke survivors, language recovery, linguistic domains, and spontaneous recovery

## Abstract

Recovery from aphasia is a complex process involving restoring language ability to a level comparable to an individual’s pre-aphasia state. This recovery extends beyond linguistic functions such as improved quality of life and functional communication. Understanding language recovery in PWAs is a key area in aphasia research. Thus, the current study aimed to understand the pattern of language recovery in the acute and sub-acute stages of persons with aphasia (PWAs). A total of 11 PWAs aged between 40 and 80 were recruited. The study was conducted in two phases. In the acute stage (within one week post-stroke), participants were assessed using the Western Aphasia Battery-Kannada (WAB-K). In the sub-acute stage (between seven and fifteen days post-stroke), a similar test battery was repeated. The findings of the study showed auditory verbal comprehension scores were pronounced in the acute and sub-acute stages of recovery. Further, language quotient (LQ) scores were higher in the sub-acute stage compared to the acute stage, though these differences failed to evince statistical differences. Correlation analysis revealed strong positive correlations between LQ and spontaneous speech, repetition, and naming, with moderate correlations for auditory verbal comprehension. The study’s findings highlight the importance of targeted therapeutic interventions for PWAs, emphasizing an early focus on auditory verbal comprehension to enhance overall language recovery.

## 1. Introduction

Aphasia, sometimes known as dysphasia, is the term used to describe the linguistic impairments that may develop following a stroke ([5]). About one-third of stroke survivors struggle with aphasia, a disorder that can impair one or more aspects of communication, including speaking, understanding spoken language, writing, and reading ([22]). The degree of this post-stroke language impairment varies and can seriously limit a person’s ability to communicate. It emphasizes how critical it is to provide specialized therapy and assistance to individuals affected by stroke to help them regain their language abilities and improve their general quality of life. Also, it is crucial to understand the nature of recovery in persons with aphasia (PWAs) across various timelines owing to its debilitating conditions.

Understanding language recovery in PWAs is one of the most intriguing topics in aphasia research ([26]). Over the past few decades, researchers have shown the nebulous nature of aphasia recovery ([2]; [8]; [17]; [14]). Despite the differences among PWAs, it is generally assumed that there is a certain degree of language improvement after a stroke. Interestingly, this improvement is even noted in PWAs who do not receive specific speech-language training ([1]; [18]; [19]) But understanding recovery patterns among PWAs remains challenging for speech-language pathologists (SLPs), owing to the influence of both neural and behavioral factors.

Recovery of aphasia is a complex process that can be understood as the restoration of language ability to a level comparable to an individual’s pre-aphasia state. This recovery does not only include restoration of linguistic functions like comprehension, expression, naming, reading, and writing but also includes other non-linguistic factors including improved quality of life and functional communication. This recovery process can vary from person to person and depends on their unique circumstances and perspective on quality of life ([7]). The process of recovery also depends on the type of aphasia as well as the area of the lesion.

Bomi [32] ([32]) conducted a study to identify the recovery process in different types of aphasia. They found that lesions in Broca’s aphasia can be related to a slower rate of recovery and when global aphasia includes lesions in the superior temporal gyrus and Broca’s area it can lead to poor prognosis. Another study conducted by [38] ([38]) posits that language difficulties have a great chance of amelioration if the lesion is circumscribed to the posterior superior temporal gyrus.

The post-stroke recovery time is mainly divided into five stages. They are hyper-acute stage (24 h), acute stage (1 to 7 days), early subacute stage (1 week–3 months), late subacute stage (3 months–6 months), chronic stage (>6 months) ([4]; [29]; [15]; [33]). The trajectory of recovery in each of these stages varies, indicating the need to understand each stage individually. For the current study, we are mainly focusing on the acute and subacute phases post-stroke to better understand and compare the language recovery in these stages.

In conclusion, stroke is a heterogeneous condition, making the prediction of outcomes and treatment strategies highly individualized. Despite this, a “one-size-fits-all” approach is often used, which negatively impacts patient recovery. Understanding the pattern of recovery, divided into five stages (hyper-acute, acute, early sub-acute, late sub-acute, and chronic), is crucial. The trajectory varies in each stage, especially in the acute and sub-acute phases, which are critical for language recovery. Accurate assessment of lesion location and volume during these phases can inform treatment and prognosis. However, challenges such as patient conditions and hospital settings complicate assessments, emphasizing the need for tailored management strategies.

### 1.1. Aphasia in the Acute Phase

The acute phase of recovery, in most cases, will be the starting point of speech and language evaluation. Several aspects emphasize the importance of assessment in the acute phase, as explained by [4] ([4]), such as lesion location and volume of the stroke; perfusion-weighted MRI in this early acute stage (within 48 h); presence of leukoaraiosis and covert lesions; measurement of the white matter integrity within both lesioned and non-lesioned areas; and prediction of the therapy responses based on lesions ([4]).

Despite the importance of assessment and rehabilitation of language in this phase, there are several challenges that an SLP will have to overcome to achieve it. At present, there are no reliable methods for predicting recovery outcomes as recovery is influenced by numerous factors, with the primary factor being the lesion site. There are several studies on discovering new methods to predict outcomes using lesion location. PLORAS system (Predicting Language Outcome and Recovery After Stroke) is one such method in which predictions are made based on data available on previous patients with the same lesion ([28]).

Along with the challenges in outcome prediction other unavoidable factors affect the assessment. Issues with consciousness and orientation, which are required for assessment to take place, and the presence of sensory difficulties may further complicate the issues. The patient might be uncooperative or temperamental ([29]).

A study by [11] ([11]) aimed to understand if some language domains assessed in the acute phase could be the prognostic indicator during aphasia recovery. Also, the study aimed to predict the pattern of aphasia through the language components in the chronic phase. The results of the study underlie the importance of assessment in the acute phase, which aids in predicting the recovery of aphasia and understanding the evolution of aphasia in the chronic phase.

The study by [16] ([16]) investigated factors such as clinical traits, outcomes, factors contributing to aphasia, and factors contributing to early improvement in acute ischemic stroke. In this study, participants who had acute ischemic stroke and were admitted to the hospital within 48 h of onset were assessed for speech and language functions on the day of admission (day 0) and day 10. The findings of the study posit that the severity of aphasia varied during the acute phase with systemic neurological severity being the only notable clinical predictor. Because predicting the outcome of aphasia in the initial days of post-stroke seems challenging, it is therefore crucial to carefully assess the need for speech therapy before initiating the treatment.

In a similar line, a study by [35] ([35]) aimed to investigate the early stages of language recovery in aphasia following stroke. Twenty-one aphasia patients were assessed over the first 15 days after an acute stroke, using the Quick Aphasia Battery. The results showed that, on average, language function improved by approximately 1.07 points per week on a 10-point scale during this period, with most patients experiencing positive changes. Recovery appeared roughly linear, with more significant improvements seen in patients with greater initial language impairment. Improvement patterns varied across language domains, with consistent progress in word finding, grammar, repetition, and reading, but less consistent gains in word comprehension and sentence comprehension. This information enhances prognosis accuracy and provides a baseline for evaluating intervention effectiveness in early aphasia recovery.

### 1.2. Aphasia in the Sub-Acute Phase

Stroke is the world’s leading cause of death and disability ([31]). Through assessment in the acute phase, imminent threats to life can be reduced, and predictions of the outcome measures can also be made. However, many stroke survivors will experience motor, sensory, perceptual, and cognitive deficits post-stroke. This will not only apply for patients but also for caretakers. Thus, rehabilitation aims to promote functional recovery and autonomy through restitution, compensation, and substitution. The focus of assessment in the sub-acute phase is mainly on monitoring the recovery process, comprehensive evaluation, rehabilitation planning, and assessing risk factors for recurrent stroke ([31]).

Similarly to the acute stage, the sub-acute stage also presents several challenges an SLP has to face during the assessment process. These can include sleep disturbances, issues with attention and memory, and various cognitive impairments ([29]). To understand the recovery process in the sub-acute phase, a better knowledge of lesion-related changes specific to stroke is needed.

Resting-state functional connectivity (rsFC) findings suggest that in the early and late sub-acute phase, interhemispheric connectivity is of importance to motor control. As several cross-sectional studies reveal that decreased rsFC correlates with the degree of motor impairment ([4]; [15]). Structural damage to the corticospinal tract (CST) was often correlated to a decrease in interhemispheric rsFC in the motor network upstream from the site of the lesion (neuroimage). Several fMRI studies in the early and late sub-acute stroke suggest that the best motor outcomes are related to the greatest shift towards the normal state and those with poorer outcomes have been suggested to have predominant inhibitory mechanisms in perilesional areas ([4]).

As explained above, the recovery pattern of each patient is different. Therefore, instead of following a common therapy pattern for all the PWAs, different intervention strategies according to the outcome measures and prognosis should be carried out. According to [31] ([31]), based on the predicted outcomes, various therapy strategies were assigned. Physiotherapy sessions address global mobility, balance, transfers, and walking. Occupational therapy sessions for training body functions such as movement, sensation, perception, and cognition as well as activities of daily living are implemented. Speech therapy sessions focus on dysphagia management, enhancing language skills, and recovering from motor speech disorder. Similarly, when focusing on language therapy if assessment has revealed the patient has issues with auditory comprehension, then other modalities could be employed for therapy. If an increased number of semantic aspects are observed, then phonemic cues can be employed in the therapy.

### 1.3. The Current Study

Most post-stroke aphasia recovery studies traditionally begin assessing patients after a delay of two weeks to one month from the onset of aphasia symptoms. This practice leaves a significant knowledge gap regarding the initial two weeks of language function development in this critical period. Some limited trials that have examined this early phase have revealed notable improvements in many patients, underscoring the importance of understanding this timeframe. A few in-depth longitudinal investigations have also documented daily language function recovery in small cohorts of one to three patients. However, these findings are challenging to generalize due to their limited sample sizes ([27]).

Along with that, it is important to comprehend how the language recuperation pattern is perceived in PWAs during the acute and sub-acute stages. This study may shed light on the elimination of some gaps that need to be filled by investigating various language domains such as spontaneous speech, auditory verbal comprehension, repetition, and naming recovering over time as well as their correlations with the general language quotient. Such findings will facilitate targeted teaching as well as therapeutic modalities for varied PWA profiles during initial post-stroke periods. It is also important for PWAs to assess predictors of aphasia improvement to make decisions about transfer to secondary institutes to receive speech and language therapy. Stroke units may need to determine whether there are indications for speech therapy within the acute phase.

The current study also serves to be essential to understanding early recovery patterns in PWAs. By comprehensively analyzing the rate of recovery across various linguistic domains, speech-language pathologists (SLPs) may provide patients and their families with a clearer understanding of what to expect during the rehabilitation journey. This can alleviate anxiety and frustration commonly experienced by both patients and caregivers. Notably, research focusing on Indian languages, specifically the Kannada-speaking population (language spoken in Karnataka state, India), is relatively scarce. Thus, the current study aimed to understand the pattern of language recovery in the acute and sub-acute stages of persons with aphasia (PWAs) in the Indian context. In this study, the following research questions were addressed:(1)Do the language domains (spontaneous speech, auditory verbal comprehension, repetition, and naming) vary in the acute and sub-acute stages of recovery?(2)Does the language quotient vary in the acute and sub-acute stages of persons with aphasia?(3)Do recovery stages (acute and sub-acute stages) have any relationship between language quotient and linguistic components (spontaneous speech, auditory verbal comprehension?

### 1.4. Methods

#### 1.4.1. Participants

In this study, a total of 11 persons with aphasia (PWAs) were recruited for the study, with their ages ranging from 40 to 80 years (M = 60.36, S. D = 14.89) (See Table 1). All these participants were recruited from Father Muller Medical Hospital, Mangalore, and Karnataka state, India. The study was approved by the Father Muller Medical College Ethics Committee (Ref No: FMIEC/CCM/015/2024; Protocol no: 023/2024). These participants recruited in the study were based on the purposive sampling method and followed the following inclusion criteria:

The participants should have been diagnosed with left Middle Cerebellar Artery (MCA) stroke; the participants should be native speakers of the Kannada language; PWAs should have post-stroke onset (PSO) time of within ‘seven’ days; the participants; and participants who have Glasgow Coma Scale (GCS; [30]) score more than ‘9’ were recruited for the study. Consequently, researchers employed the following exclusion criteria to rule out the participants from the study. PWAs who had stroke post-onset (SPO) period of more than one month were excluded; PWAs who had aphasia with dementia were excluded using MOCA ([24]); PWAs with serious mental illnesses or psychiatric conditions that were gauged via taking detailed medical history were excluded; and PWAs with associated problems leading to stroke, such as tumors, rraumatic brain injury (TBI), or infectious diseases, were excluded.

#### 1.4.2. Sample Size Estimation

Sample size was calculated using the following formula n = (Zα + Zβ)^2^σ^2^/(x_1_ − x_2_)^2^ Where Zα = 1.96, at 95% confidence interval, Zβ = 1.21, at 90% power, σ = 1.5, polled standard deviation, (x_1_ − x_2_) = 1.5, mean difference. The sample size was estimated to be ‘11’ at 90% power.

#### 1.4.3. Variables Considered in the Study

The dependent variables considered in the study were spontaneous speech, auditory verbal comprehension, repetition, naming, and language scores. The independent variable considered in this study was Bedside Western Aphasia Battery-Kannada (Bedside WAB-K).

### 1.5. Procedure

The study used in the study was Bedside Western Aphasia Battery-Kannada (Bedside WAB-K) ([6]). Normative & Clinical Data on the Kannada Version of Western Aphasia Battery (WAB-K). It has ‘four’ linguistic domains, which were chosen from the sub-categories of the Bedside WAB-K. The test is a well-established tool in language assessment, providing a reliable framework for analyzing linguistic abilities. By utilizing its sub-categories, the study aimed to capture a detailed and nuanced understanding of language performance. The scoring of all the sections pertaining to the test is performed according to the standardized norms prescribed in the original test. The test categories assessed were mentioned below:

#### 1.5.1. Spontaneous Speech

Each participant engaged in a general discussion in which their fluency and content were rated. The participant’s communication modality considered was verbal.

#### 1.5.2. Auditory Verbal Comprehension

(a)Yes/No questions. The participant was asked a series of questions, such as “Are you playing now?” and expected to respond with either ‘yes’ or ‘no’.(b)Pointing task. Picture cards were kept in front of each participant. Then the examiner showed picture cards to the participants, and they had to point to the one the examiner told them about. For example, if someone says, “Point to the leg,” the response should be “Person pointing to the leg”.(c)Auditory Word Recognition. The examiner gave words in order, and the participants were required to recognize them and respond by moving their eyes toward the object, making a gesture, or pointing to it. For example, if the examiner says “window,” the reaction can be “looking towards the window,” “pointing to it,” or any other “gestural response” indicating that the subject recognizes the term.(d)Verification task. For this subtest, a single card with three photographs was displayed. The photographs were shown sequentially, in the same order as the test stimuli. Then the instructions were to thoroughly inspect the picture and point to the one that the examiner requested. For example, if the examiner wants to see ‘banana,’ the response should point to ‘banana’ by confirming the images shown.(e)Sequential commands. On the table, a set of objects was arranged one after the other. Then the participants were instructed to follow the commands and perform the actions with objects that were placed in front of them. The examiner reminds them, whenever needed, to perform the actions sequentially according to the prior instructions; e.g., if the examiner says, “Close your eyes” the response should be “Closing his/her eyes”.

#### 1.5.3. Repetition

(a)Automatic Speech. Participants were instructed by the examiner to answer a series of ordered questions. For example, if the instruction was to “count from ‘one’ to ‘ten’,” the expected response from the participants would be reciting the numbers in sequence from ‘one’ to ‘ten’.(b)Word. Participants were asked to repeat specific words spoken by the examiner. For example, if the examiner said, “book,” the participant was required to simply repeat the word “book.”(c)Phrase. Participants were instructed to repeat phrases after the examiner. For instance, if the examiner said, “come here,” the participant needed to repeat the exact phrase “come here.”(d)Sentence. Participants were given instructions to repeat sentences after the examiner. For example, if the examiner said, “I want this bag,” the participant had to repeat the sentence “I want this bag” exactly as stated.

#### 1.5.4. Naming

(a)Confrontation Naming. On the table, the picture cards were presented in the order of the presentation. Then the instruction given was to name the picture that was presented by the examiner. For example, if the picture presented was “spoon”, the participant should name it “spoon”.(b)Responsive Naming. The participant was asked to answer the questions by naming the question; e.g., if it was “What can you see in the kitchen”, the participant should name “All possible items that he/she can see in the kitchen”.(c)Lexical Generative Naming. The participant was instructed to name the things that come under one specific category that were told by the examiner. For example, if it was “list the animals”, the participants should name “All possible animals he/she can”.

Further, the study was conducted in two phases, acute and sub-acute phases of assessment:*Phase 1—The Acute-Phase Assessment*

The acute-phase assessment took place within one week following the onset of stroke ([4]; [29]; [15]; [33]). In this phase, MoCA, GCS, and Bedside WAB-K were administered. The aforementioned procedures of WAB-K administration are elucidated.


*Phase 2—The Sub-Acute-Phase Assessment*


The sub-acute-phase assessment was conducted after one week post-stroke ([4]; [29]; [15]; [33]). This phase of assessment is expected to be completed within ‘seven’ days to ‘fifteen’ days. In this phase, the researcher repeated the administration of Beside WAB-K. The procedure for this phase is akin to that of the acute phase assessment except for the administration of MoCA and GCS.

### 1.6. Data Analysis

Analyzed data were tabulated and subjected to appropriate statistical analysis using SPSS software version 21.0. Initially, data was subjected to a normality check using the Shapiro–Wilk test, owing to a non-normal distribution of the data (*p* < 0.05). Non-parametric test was used to see the statistical differences. The scoring of each section is as per the standard norm of the WAB-K.

## 2. Results

### 2.1. Comparing Language Domains (Spontaneous Speech, Auditory Verbal Comprehension, Repetition, and Naming) Within the Acute and Sub-Acute Stages of Recovery

Descriptive analysis for the acute phase revealed that median scores of auditory verbal comprehension were higher followed by spontaneous speech, repetition, and naming (See Figure 1).

Further, these measures were subjected to statistical analysis using the Friedman two-way test. This test was performed to ascertain the differences among the language domains that were analyzed (i.e., spontaneous speech, auditory verbal comprehension, repetition, and naming). The results revealed that there is a significant difference among the language domains in the acute stage of recovery (χ^2^ (4) = 31.17, *p* < 0.05). Consequently, the results were subjected to a pairwise difference using the Wilcoxen sign rank test in the acute stage. The results revealed a significant difference in auditory verbal comprehension versus spontaneous speech; repetition versus auditory verbal comprehension; and naming versus auditory verbal comprehension (See Table 2).

The descriptive analysis for the sub-acute phase showed that the median score of auditory verbal comprehension was pronounced in the sub-acute stage of recovery, followed by spontaneous speech, repetition, and naming (See Figure 1). Like the acute stage, these measures were analyzed using the Friedman two-way test to determine differences among the language domains (spontaneous speech, auditory verbal comprehension, repetition, and naming). The results indicated a significant difference among the sub-acute stage of recovery (χ^2^ (4) = 33.23, *p* < 0.05). Subsequently, the Wilcoxon sign rank test was conducted to identify pairwise differences in the sub-acute stage. The results showed significant differences in the following pairs: auditory verbal comprehension versus spontaneous speech; repetition versus auditory verbal comprehension; and naming versus auditory verbal comprehension (See Table 3).

### 2.2. Comparing Language Quotient Across the Acute and Sub-Acute Stages of Persons with Aphasia

Descriptive evaluation revealed median score for language quotient was higher in a sub-acute stage than of acute stage (See Table 4).

Furthermore, a pairwise comparison was conducted using the Wilcoxon sign rank test among the language quotients of the acute and sub-acute stages. Results revealed no significant difference in the acute stage and the sub-acute stage of recovery (|Z| = 2.00, *p* = 0.45).

### 2.3. Correlation Between Language Quotient and Linguistic Components (Spontaneous Speech, Auditory Verbal Comprehension, Repetition, and Naming) in Acute and Sub-Acute Stages of Recovery

Pearson’s correlation analysis was performed to see the correlation between language quotient and language domains. In the acute phase, spontaneous speech and language quotient have a very strong positive correlation (*r* = 0.95, *p* < 0.05). Auditory verbal comprehension and language quotient has a moderate positive correlation (*r* = 0.53, *p* < 0.05). Repetition and language quotient has a very strong positive correlation (*r* = 0.77, *p* < 0.05). Naming and language quotient has a very strong positive correlation (*r* = 0.81, *p* < 0.05).

In the sub-acute phase, results revealed a very strong and highly significant positive correlation between spontaneous speech and language quotient (*r* = 0.97, *p* < 0.05), a moderate positive correlation was observed between auditory verbal comprehension and language quotient (*r* = 0.56, *p* < 0.05), a strong and statistically significant positive correlation between repetition and language quotient (*r* = 0.82, *p* < 0.05) and a very strong and highly significant positive correlation was noted between naming and language quotient (*r* = 0.86, *p* < 0.05).

To summarize, the study examined language recovery in PWAs during the acute and sub-acute stages. It compared the language domains and their components (spontaneous speech, auditory verbal comprehension, repetition, and naming), with the findings showing there is a significant difference among these categories in both stages. In the acute phase, auditory verbal comprehension had the highest median, followed by spontaneous speech, repetition, and naming. The sub-acute stage also showed similar patterns. The language quotient was higher in the sub-acute stage compared to the acute stage; despite the differences in median scores, the researcher failed to provide evidence of statistical differences in the language quotient. Pearson’s correlation revealed strong positive correlations among language quotient- spontaneous speech, repetition, and naming. However, the researcher found a moderately positive correlation between language quotient and auditory verbal comprehension. A similar trend was also observed in the sub-acute stage of recovery.

## 3. Discussion

The overarching aim of this study was to understand the pattern of aphasia recovery in the acute and sub-acute stages. The overall results of the study demonstrated a notable improvement in the recovery of auditory verbal comprehension at a significantly higher rate compared to the other assessed categories. In the first objective, the goal was to compare different language domains—spontaneous speech, auditory verbal comprehension, repetition, and naming within acute and sub-acute stages of recovery. The findings revealed that auditory verbal comprehension scored significantly higher compared to the other three assessed domains. This observation aligns with multiple studies indicating that the recovery rate of auditory verbal comprehension tends to be faster compared to other linguistic categories ([21]).

According to the literature, several factors may influence the overall cognitive and functional recovery in stroke patients. These factors include social support ([13]; [25]) psychiatric sequelae ([20]), access to high-quality medical treatment and fewer comorbidities ([23]; [36]), and engagement in regular exercise. It is plausible that these elements contributed to the spontaneous improvement in auditory comprehension in the current study. However, further investigation is required to explore this possibility comprehensively.

Another plausible explanation for the higher scores in auditory verbal comprehension could be related to the participant selection process. During analysis, participants were not categorized based on the type of aphasia. As a result, there has been a higher proportion of participants diagnosed with non-fluent aphasia, a condition in which comprehension abilities are typically more preserved. This potential skew in the participant pool could have influenced the overall results, emphasizing the need for more stratified sampling in future studies to better understand the nuances of recovery across different types of aphasia.

The second objective was to compare the language quotient across acute and sub-acute stages of recovery. No significant changes were observed among the language quotient in both phases. Various factors could contribute to the lack of improvement and reduce the patient’s cognitive state. Decreased communication participation over time ([34]). Decreased integration of sensory inputs (hearing and vision) ([3]), or inconsistent routines and lack of activities ([34]).

Other reasons specific to this study that may have led to the lack of significant changes from the acute to the sub-acute stage could include the short interval between the two assessments. The brief gap may not have provided enough time to observe significant changes. The standardized test used in the study may not be subtle enough to gauge the linguistic improvement in these individuals.

Other reasons like the severity of aphasia that may impact the rate of recovery, differences in the intensity and type of rehabilitation interventions received by the patients, or even psychological factors such as motivation and mood can all impact recovery and assessment outcomes. Even heterogeneity of the patient populations may also mask the overall trends.

The final objective of this study was to identify the correlation between the language quotient (LQ) and various language domains during the acute and sub-acute stages of recovery, focusing on auditory verbal comprehension, spontaneous speech, repetition, and naming. The results revealed a positive correlation across all four assessed categories. When each category is considered separately, spontaneous speech exhibited a strong positive correlation with LQ, suggesting that patients with higher LQ scores are likely to demonstrate better spontaneous speech capabilities. This finding aligns with the notion that overall language proficiency underpins the ability to produce spontaneous speech, corroborating prior studies ([12]) that reported significant relationships between linguistic abilities and spontaneous speech in aphasia.

Similarly, auditory verbal comprehension showed a positive correlation with LQ. The high rate of recovery in auditory verbal comprehension contributed to the improvement of the language quotient, which is consistent with several studies ([9]). Repetition also exhibited a strong positive correlation with LQ. This correlation indicates that LQ, encompassing a range of linguistic abilities such as vocabulary, grammar, and comprehension, is a significant predictor of repetition recovery.

Furthermore, this study identified a strong positive correlation between LQ and naming recovery in PWAs. Higher LQ scores were associated with better recovery of naming abilities, suggesting that overall language proficiency is a crucial factor in the rehabilitation of naming functions in PWAs. These findings are consistent with other studies that have highlighted similar correlations ([37]; [9]).

The current study’s findings highlight the importance of considering the differential recovery rates of various language domains in aphasia rehabilitation. The faster recovery rate of auditory verbal comprehension appears to play a crucial role in the overall improvement of the language quotient. This underscores the need for targeted therapeutic interventions that address the specific needs of each linguistic component. Further research is essential to explore these correlations in greater detail and to develop more nuanced rehabilitation strategies tailored to the unique recovery patterns of stroke patients.

## 4. Conclusions

Overall, this study highlights the significance of understanding acute and sub-acute phases of recovery. The findings of the study highlight the fact that auditory verbal comprehension (AVC) scores showed robust effects compared to others in the initial stages of recovery, thus focusing on AVC can enhance other language domains. It also underscores the importance of a comprehensive assessment that addresses various language areas. Furthermore, therapy planning should be tailored specifically to each individual. By prioritizing auditory verbal comprehension early in recovery, we can promote improvements across different language domains, making personalized therapy and comprehensive assessments crucial for effective recovery and overall language development.

### Implications

Understanding the differential recovery patterns across language domains can inform targeted therapeutic interventions, prioritizing domains like auditory verbal comprehension in the early stages to maximize overall language recovery. The strong correlations between LQ and language domains further emphasize the need for a holistic approach to aphasia rehabilitation, where improvements in individual language skills are viewed as interconnected and mutually reinforcing.

Future research could explore the underlying neural mechanisms driving the observed recovery patterns, potentially integrating neuroimaging techniques to correlate brain activity with language recovery trajectories. Additionally, longitudinal studies extending beyond the sub-acute stage could provide a more comprehensive understanding of long-term recovery patterns and the sustainability of improvements across different language domains. Also, the study can be extended by observing one variant of aphasia over a period of time, which yields better insights in understanding the recovery of PWAs.

## 5. Limitations

The current study which intends to study the pattern of recovery in acute and sub-acute stages of PWAs has several setbacks. Firstly, the small sample size limits the generalizability and robustness of the findings. Additionally, the focus on only the acute and sub-acute stages excludes insights into longer-term recovery patterns, which are crucial for understanding the full trajectory of language recovery. The absence of a control group and potential bias in subjective language assessments further constrain the study’s ability to isolate the effects of aphasia and ensure objective evaluations. Variability in aphasia severity among participants was not accounted for, potentially affecting the observed recovery patterns. Moreover, the study’s limited focus on specific language domains (spontaneous speech, auditory verbal comprehension, repetition, and naming) overlooks other important language domains like reading and writing. The variability in rehabilitation interventions among participants might have influenced outcomes. Lastly, the focus on language recovery without assessing functional impacts on daily living and quality of life leaves out important dimensions of recovery that are critical for a holistic understanding.

## Figures and Tables

**Figure 1 behavsci-15-01339-f001:**
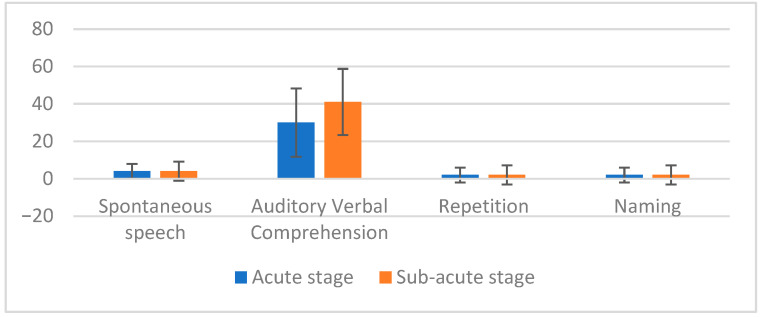
Descriptive analysis of acute and sub-acute stages.

**Table 1 behavsci-15-01339-t001:** Demographic details of the participants.

Sl. No	Age/Sex	Lesion Sites	Occupation	Post-Stroke Onset
1	40/M	Left frontoparietal infarct	Government employee	3 days
2	80/M	Left frontoparietal infarct	Retired	1 day
3	48/M	Left temporal lobe infract	Business	4 days
4	80/F	Left temporoparietal infarct	Retired	2 days
5	41/F	Left MCA anterior division territory infarct	Teacher	2 days
6	59/M	Left temporal lobe infarct	Business	6 days
7	77/M	Left parietal lobe infarct	Retired	5 days
8	59/F	Left frontoparietal temporal region and corona radiata infarct	Housewife	5 days
9	64/F	Left MCA territory to infarct	Retired	5 days
10	48/F	Left parietal lobe infarct	Government employee	4 days
11	68/F	Left frontoparietal infarct	Retired	6 days

**Table 2 behavsci-15-01339-t002:** Pairwise comparison across language domains in the acute stage.

Pairs	|Z|-Value	*p*-Value	Effect Size (re)
AVC vs. SS	2.66	0.008 *	0.80
Rep vs. SS	0.30	0.759	0.09
Naming vs. SS	1.54	0.122	0.46
Rep vs. AVC	2.80	0.005 *	0.84
Naming vs. AVC	2.80	0.005 *	0.84
Naming vs. Rep	1.57	0.115	0.47

Note * indicates significant difference; AVC = Auditory Verbal Comprehension, SS = Spontaneous Speech, Rep = Repetition; Effect size (re) if <0.3 was considered as low, 0.3–0.5 was considered as medium, and >0.5 was considered high ([10]).

**Table 3 behavsci-15-01339-t003:** Pairwise comparison across language domain in the sub-acute stage.

Pairs	|Z|-Value	*p*-Value	Effect Size (re)
AVC vs. SS	2.80	0.005 *	0.84
Rep vs. SS	0.25	0.798	0.07
Naming vs. SS	0.77	0.441	0.23
Rep vs. AVC	2.93	0.003 *	0.88
Naming vs. AVC	2.93	0.003 *	0.88
Naming vs. Rep	1.36	0.173	0.41

Note * indicates significant difference; AVC = Auditory Verbal Comprehension, SS = Spontaneous Speech, Rep = Repetition; Effect size (re) if <0.3 was considered as low, 0.3–0.5 was considered as medium, and >0.5 was considered high ([10]).

**Table 4 behavsci-15-01339-t004:** Language quotient scores across acute and sub-acute phases.

Conditions	Median	Standard Deviation	Quartiles		
			25th	50th	75th
Acute phase	10.70	11.05	6.80	10.70	11.30
Sub-acute phase	11.30	14.30	3.60	11.30	23.50

## Data Availability

The datasets analyzed in this study are available in https://docs.google.com/spreadsheets/d/1tMRpYLO616feL0R-2fWmwJjIr2-fRXq5/edit?gid=1460743552#gid=1460743552, accessed on 15 September 2025.

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
