# Peer review of "Exploring Language Recovery Pattern in Persons with Aphasia Across Acute and Sub-Acute Stages"

_behavsci, 2025, doi:10.3390/bs15101339_

Round 1

Reviewer 1 Report

Comments and Suggestions for Authors

This manuscript investigates changes in language abilities during the first two weeks following stroke, specifically assessing spontaneous speech, auditory verbal comprehension, repetition, and naming with the Kannada version of the Western Aphasia Battery (WAB-K). While the topic is clinically valuable and timely, several points need addressing to clarify findings and strengthen their scientific impact. Please see my detailed comments below.

Major comments:
1.  The authors define the acute stage as ≤7 days and the subacute stage as 7-15 days post-stroke. However, standard definitions typically label 7-15 days as an "early subacute" rather than a separate subacute phase (Boyd et al., 2017). Please clarify your rationale and consider relabelling the second period explicitly as the "early subacute" phase.

2. The sample size (n=11) is small, and the provided power calculation is conducted post hoc using the same sample size, leading to circular logic. It would be helpful to clarify whether a prospective sample-size calculation was registered in advance. Additionally, please temper statements regarding generalisability throughout the manuscript given the small sample size.

3. Participants' lesion sites are varied (left temporal, parietal, frontoparietal), but no details on aphasia severity, lesion volume, or prior therapy exposure are provided (see Table 1). Please report these clinical covariates or explicitly discuss how their omission may impact the interpretation of language domain recovery.

4. Data were not normally distributed, as indicated by the significant Shapiro-Wilk tests, yet parametric Pearson correlations were used. Please replace Pearson correlations with Spearman's correlations or justify clearly why Pearson's method was selected despite violations of statistical assumptions.

5. The manuscript reports statistically significant differences among language domains at each assessment (Tables 2 and 3), but the overall Language Quotient (LQ) showed no significant change between acute and subacute stages (Wilcoxon test: p=0.45). Despite this, the authors strongly advocate for prioritizing auditory verbal comprehension therapy early on (see abstract and discussion conclusions). Please revise this recommendation to accurately reflect the modest statistical findings.

6. Two recent systematic reviews (Wilson et al., 2019; Wilson & Schneck, 2021) provide relevant context on early domain-specific recovery and neural plasticity. Please refer to these when framing your findings, and comment briefly on whether your data support or extend these prior reports.

Minor comments:

1. Figure 1 is currently unclear: axes labels and units are missing. Please improve readability by clearly labeling axes.
2. Standardise terminology throughout: consistently use either "subacute" or "sub-acute," not both.

Author Response

Reviewer Comments (RC)-The authors define the acute stage as ≤7 days and the subacute stage as 7-15 days post-stroke. However, standard definitions typically label 7-15 days as an "early subacute" rather than a separate subacute phase (Boyd et al., 2017). Please clarify your rationale and consider relabeling the second period explicitly as the "early subacute" phase.

Author response (AR)- Thank you for the comment. In line with Boyd et al. (2017), we acknowledge that the 7–15-day period is more accurately described as the "early subacute" phase rather than a distinct "subacute" phase. Hence, we have revised the term subacute phase to early subacute phase. Rational for including these phases is provided in the section of ‘the current study’ (Introduction section). 

RC-The sample size (n=11) is small, and the provided power calculation is conducted post hoc using the same sample size, leading to circular logic. It would be helpful to clarify whether a prospective sample-size calculation was registered in advance. Additionally, please temper statements regarding generalisability throughout the manuscript given the small sample size.

AR- Thank you for the comment.

 We acknowledge that, although the Language Quotient (LQ) did not show a statistically significant difference between the acute and early subacute stages (Wilcoxon test: p = 0.45), individual language domain scores—particularly auditory verbal comprehension—demonstrated statistically significant improvements during both phases (as shown in Tables 2 and 3).

Our recommendation to prioritize auditory verbal comprehension in early therapy was based on these domain-specific findings, as well as the strong positive correlation observed between LQ and auditory verbal comprehension across both stages. However, we agree with the reviewer that this recommendation should be more cautiously stated to align with the modest overall statistical findings.

Accordingly, we have revised the abstract, discussion, and conclusion sections to temper our recommendation. We now emphasize that auditory verbal comprehension may serve as a promising early therapeutic focus based on its relatively strong recovery trend, but that this suggestion should be interpreted in the context of the preliminary nature of the study, small sample size, and the non-significant change in overall LQ. (See Discussion section) 

.RC- Two recent systematic reviews (Wilson et al., 2019; Wilson & Schneck, 2020) provide relevant context on early domain-specific recovery and neural plasticity. Please refer to these when framing your findings, and comment briefly on whether your data supports or extend these prior reports. 

. AR-Thank you for the comment. Changes have been incorporated in the manuscript , See introduction and discussion section. 

RC-Figure 1 is currently unclear: axes labels and units are missing. Please improve readability by clearly labeling axes. 

AR- Thank you for the comment. Changes have been incorporated in the manuscript

RC-Standardize terminology throughout: consistently use either "subacute" or "sub-acute," not both.

AR-Thank you for the comment. We have used the phrase subacute throughout the manuscript.

Reviewer 2 Report

Comments and Suggestions for Authors

The authors studied the linguistic performance of 11 participants with vascular aphasia in the acute and subacute stages. The topic is interesting, but the sample is too small and heterogeneous to warrant a statistical analysis. I believe that it might be best to present descriptive findings only, while some additional comments are required.

Throughout the text, the authors refer to the outcomes of therapeutic interventions as consequences of patient assessment in several different instances. This is not a widespread notion, since the reliability of therapeutic interventions for language recovery is questionable at best. There is some evidence that speech rehabilitation has some success for patients with dysarthria only. In addition, this study does not seem to have included therapeutic interventions as variables to the outcomes.

In the abstract, the conclusion that "The study's findings highlight the importance of targeted therapeutic interventions for PWAs, emphasizing an early focus on auditory verbal comprehension to enhance overall language recovery" can not be drawn from the findings of the study. The study comprised only language assessments, not interventions.

In the Introduction, the sentence "It emphasizes how critical it is to provide specialized therapy and assistance to individuals affected by stroke to help them regain their language abilities and improve their general quality of life" requires a reference. In addition, the Introduction as a whole is too long, and could be considerably shortened by the removal of paragraphs referring to small studies with inconsistent conclusions.

The first paragraph of the section "Aphasia in the acute phase" is incomprehensible, and must be rewritten for clarity. Furthermore, concerning the sentence "At present, there are no reliable methods for predicting recovery outcomes as recovery is influenced by numerous factors, with the primary factor being the lesion site", I am not aware of lesion site being the primary factor, also because no references were presented for this sentence.

Concerning the exclusion criterion "PWAs who had aphasia with dementia were excluded using MOCA", how was it possible to use the MoCA to exclude dementia? The Montreal Cognitive Assessment is a screening cognitive test only, and unable to diagnose a dementia syndrome.

Prognostic factors for language recovery after a vascular aphasic syndrome is diagnosed include age, education, depression, lesion size, lesions in the dominant hemisphere and cortical injuries (see Brain Injury 31(2):140-150, https://doi.org/10.1080/02699052.2016.1199914 where potential reasons are discussed). The authors do not seem to have taken these prognostic indicators into account in their study, so a comment is required.

In the Discussion, I could not understand how the authors reached the following conclusion: "It is plausible that these elements contributed to the spontaneous improvement in auditory comprehension in the current study".

The conclusion "By prioritizing auditory verbal comprehension early in recovery, we can promote improvements across different language domains, making personalized therapy and  comprehensive assessments crucial for effective recovery and overall language development" can not be drawn from the findings of the study.

Comments on the Quality of English Language

The article requires careful proofreading. Several misconstrued sentences may be found all along the text.

Author Response

Reviewer Comments (RC): The authors studied the linguistic performance of 11 participants with vascular aphasia in the acute and subacute stages. The topic is interesting, but the sample is too small and heterogeneous to warrant a statistical analysis. I believe that it might be best to present descriptive findings only, while some additional comments are required. 

Author Reply (AR)- 

In our current study, while we documented participants’ age and lesion sites descriptively, we did not perform a detailed stratified analysis of these prognostic indicators due to the small sample size and exploratory nature of the study.

Moreover, variables such as education level and depression history were not systematically controlled or assessed, which we now recognize as a limitation. We have added a statement to the Limitations section to reflect this and have emphasized that future studies with larger and more controlled cohorts should incorporate these factors to better account for their influence on recovery patterns.

RC-In the Discussion, I could not understand how the authors reached the following conclusion: "It is plausible that these elements contributed to the spontaneous improvement in auditory comprehension in the current study".

AR-Thank you for the comment. We have reached to this conclusion firstly based on ancetotal reports, stating all participants participated in the study had good social support, motivated, and was getting good medical intervention, thus, authors claim that improvement could be due to this factors. However, authors also cautioned researchers and readers, that further investigation is required to explore this possibility comprehensively. 

RC-The conclusion "By prioritizing auditory verbal comprehension early in recovery, we can promote improvements across different language domains, making personalized therapy and  comprehensive assessments crucial for effective recovery and overall language development" can not be drawn from the findings of the study. 

AR- We thank the reviewer for this important observation. We agree that the original conclusion may have unintentionally conveyed a stronger causal inference than our data can support.

Our intent was not to assert that prioritizing auditory verbal comprehension will directly lead to improvements in other language domains, but rather to suggest that the relatively stronger early recovery of auditory comprehension observed in our participants may indicate its potential as a starting point or early focus in therapy planning.

To address this, we have revised the relevant sentence in the Discussion and Conclusion sections to more accurately reflect the exploratory and observational nature of our findings.

Round 2

Reviewer 2 Report

Comments and Suggestions for Authors

Most of my comments were inappropriately addressed:

  1. The Introduction is excessively long, spanning 9 pages. The authors seem undecided on whether they are going to write a review article or introduce a new original article.
  2. The sample is overanalyzed by the excessive statistics. A descriptive study would be more suitable for such a small sample.
  3. Many sentences remain unsupported by proper references.
  4. The authors now write that "PWAs exhibiting aphasia accompanied by cognitive impairment were excluded based on the Montreal Cognitive Assessment" - how was the MoCA used to exclude cognitive impairment?
  5. The following conclusion has not been explained in the text: "It is plausible that these elements contributed to the spontaneous improvement in auditory comprehension in the current study". In my view, it is impossible to reach this conclusion from the findings of the study.
Comments on the Quality of English Language

The article could still benefit from careful proofreading.

Author Response

Reviewer Comment (RC): The Introduction is excessively long, spanning 9 pages. The authors seem undecided on whether they are going to write a review article or introduce a new original article. 

Author Response (AR)-Thank you for the comment; the introduction section has been condensed from 9 pages to 4 pages. 

RC-The sample is overanalyzed by the excessive statistics. A descriptive study would be more suitable for such a small sample.

AR-Thank you for your valuable comment.

We acknowledge that the sample size in the current study is relatively small, despite being guided by appropriate sample size estimation methods. This limitation has been clearly stated in the manuscript.

However, we would like to highlight that several studies in the field of aphasiology with similarly small sample sizes have successfully employed statistical analyses to compare different stages of recovery, such as subacute and chronic aphasia. These studies have been accepted in reputable, peer-reviewed journals (e.g., (e.g., Bomi Sul et al., 2016;https://doi.org/10.5535/arm.2016.40.5.786

Mackenize et al., 2016; https://doi.org/10.1080/10749357.2016.1155277

We have taken care to present our findings as preliminary and have refrained from overgeneralization. Our intent is to contribute early insights that may inform future research with larger, more representative samples.

Additionally, in the results section, we have first presented a descriptive analysis of the data to provide a clear overview of trends and patterns, followed by inferential statistical tests to explore the presence or absence of significant differences. We believe this structured approach offers a balanced interpretation of the data within the constraints of the sample size.

We appreciate your input and will continue to be mindful of these considerations in future work.

RC-Many sentences remain unsupported by proper references. 

AR-Thank you for your comment. The missing sentences have been backed up by appropriate references, and changes have been highlighted. 

RC-The authors now write that "PWAs exhibiting aphasia accompanied by cognitive impairment were excluded based on the Montreal Cognitive Assessment" - how was the MoCA used to exclude cognitive impairment?

AR-

Thank you for the comment.

The Montreal Cognitive Assessment (MoCA) is a widely used, brief cognitive screening tool designed to detect mild cognitive impairment, as originally described by Nasreddine et al. (2005) [https://doi.org/10.1111/j.1532-5415.2005.53221.x]. In our study, the MoCA was administered to screen participants for cognitive deficits beyond language impairment that might confound the results. 

While we recognize that there are articles that they have used

used MoCA to screen cognitive impairment in aphasia (e.g., Annemarie et al., 2023; https://doi.org/10.1080/02699052.2024.2341039

Considering the nature of the test and based on previous studies, we have used MoCA to screen for cognitive impairment in our study.

RC-

The following conclusion has not been explained in the text: "It is plausible that these elements contributed to the spontaneous improvement in auditory comprehension in the current study". In my view, it is impossible to reach this conclusion from the findings of the study.

AR-

Thank you for your comment.

We have changed the discussion points explaining natural recovery mechanisms during acute phases, and the detailed explanation is given in the discussion section. 
